# Surface and Tribological Properties of Powder Metallurgical Cp-Ti Titanium Alloy Modified by Shot Peening

Yasemin Yıldıran Avcu [1], Eleftherios Iakovakis [2], Mert Guney [3,4,*], Emirhan Çalım [1], Ayşe Özkılınç [1], Eray Abakay [5], Fikret Sönmez [6], Funda Gül Koç [7], Rıdvan Yamanoğlu [7], Abdulkadir Cengiz [8] and Egemen Avcu [1,9,*]

1 Department of Mechanical Engineering, Kocaeli University, Kocaeli 41001, Turkey
2 Department of Mechanical Aerospace and Civil Engineering, The University of Manchester, Manchester M13 9PL, UK
3 Department of Civil and Environmental Engineering, Nazarbayev University, Astana 010000, Kazakhstan
4 The Environment & Resource Efficiency Cluster (EREC), Nazarbayev University, Astana 010000, Kazakhstan
5 Department of Metallurgy and Materials Engineering, Sakarya University, Sakarya 54050, Turkey
6 Hasan Ferdi Turgutlu Faculty of Technology, Department of Mechanical Engineering, Manisa Celal Bayar University, Manisa 45140, Turkey
7 Department of Metallurgy and Materials Engineering, Kocaeli University, Kocaeli 41001, Turkey
8 Department of Automotive Engineering, Kocaeli University, Kocaeli 41001, Turkey
9 Ford Otosan Ihsaniye Automotive Vocational School, Kocaeli University, Kocaeli 41650, Turkey
* Correspondence: mert.guney@nu.edu.kz (M.G.); avcuegemen@gmail.com (E.A.);
  Tel.: +7-71-7270-4553 (M.G.); +90-26-2435-6167 (E.A.)

**Abstract:** The present study reveals for the first time the dry sliding wear behavior of a powder metallurgical pure titanium alloy (Cp-Ti) modified by shot peening. Cp-Ti samples were manufactured via powder metallurgy, and then their surface and subsurface features were modified using a custom-made, fully automated shot-peening system. The texture isotropy rate and the highest orientation angle of the shot-peened samples were 71.5% and 36°, respectively. The Abbott curves of the shot-peened surfaces revealed that the most common areal roughness value was 5.177 μm, with a frequency of 8.1%. Shot-peened surfaces exhibited an ~20% lower wear rate than unpeened surfaces under dry sliding wear, whereas the coefficient of friction was the same for both surfaces. Micro-ploughing, micro-cutting, oxidation, and three-body abrasion wear mechanisms were observed on the shot-peened and unpeened surfaces. High resolution 3D surface topographies of worn unpeened and shot-peened surfaces revealed micro-scratches and inhomogeneities along wear tracks, which are indicative of three-body abrasion mechanisms during contact. In addition, vertical and horizontal microcracks were visible just beneath the wear track, suggesting a clear indication of plastic deformation during contact. The cross-sectional hardness maps of shot-peened samples revealed the formation of a work-hardened surface layer with shot peening, which improved the wear resistance. These findings support that shot peening can be a useful tool to modify the surface and tribological properties of powder metallurgical Cp-Ti alloys.

**Keywords:** friction; hardness mapping; implants; shot peening; titanium alloys; topography; wear rate

## 1. Introduction

Titanium (Ti) and its alloys have been widely used in numerous engineering applications (e.g., aerospace, biomedical) owing to their superior specific strength [1], corrosion resistance [2], and biocompatibility [3,4]. Among Ti alloys, commercially pure Ti (Cp-Ti) is prominent for medical and surgical uses in applications such as bio-implantable bone substitutes due to its biocompatibility, mechanical behavior, and corrosion resistance [5,6]. It is often used in dental implants, favored by its desired biological properties coupled with low Young's modulus and suitable strength [3]. However, its relatively poor tribological properties (i.e., coefficient of friction (CoF), wear rate) restrict its use in applications where

sliding, fretting, and rolling contact are inevitable [2–4]. Its relatively poor tribological properties [3] are mostly related to its plastic deformability and weak work hardening [4].

To date, various surface treatments, including plasma electrolytic oxidation [7], anodization [8], large-strain extrusion processing [9], high-temperature oxidation [3], shot peening [10], laser peening [11], and surface mechanical attrition treatment (SMAT) [6], have been used to improve the tribological properties of Ti alloys. Among these methods, SMAT [6,12,13] and shot peening [14,15] have been the most common mechanical surface treatment methods. Fundamentally, both methods modify the surface and subsurface microstructural and mechanical properties by plastically deforming the region of interest [6,12–15]. However, shot peening has several significant advantages over SMAT, including simple application, applicability to components with complex shapes, and low process cost [14].

Recently, the number of studies exploring the effects of shot peening on the tribological properties of engineering materials has increased [16–30] since shot peening has the potential to improve the tribological properties of materials along with its well-known effect of enhancing the fatigue strength and service life of engineering materials. Shot peening modifies the surface and subsurface properties of materials via the repeated impact of high-velocity shots accelerated through a nozzle on their surfaces (e.g., hardness, residual stress, microstructural features) [14,15,31], which can also affect their tribological properties. For instance, Han et al. [21] reported a decrease in the wear resistance of AIS 5160 steel after shot peening, which was attributed to the formation of cracks and increased surface roughness with shot peening. In contrast, Zhan et al. [28] reported enhanced wear resistance in S30432 stainless steel due to increased surface hardness and obtained gradient hardening layers via shot peening. Increased wear resistance has also been reported for 17-4PH stainless steel [25], 4Cr9Si2 valve steel [22], 17Cr2Ni2MoVNb steel [30], AISI 5160 steel [20], and C45 steel [27] associated with the following modifications: increased modification of surface topography reducing local stress, storing wear debris and improving lubrication during contact, and synergistic effects of grain refinement and hardness. Nevertheless, most of these studies have been conducted on steel samples, whereas studies on Ti alloys have remained limited [17,26,32].

DiCecco et al. [17] found that the CoF and wear rate of shot-peened alpha Ti alloy were not significantly different from those of unpeened alloy samples under dry sliding wear conditions. In contrast, Ti6Al4V alloy exhibited improved tribological performance in wear tests in Ringer solution [32], attributable to increased hardness with shot peening. Yang et al. [26] reported that shot peening altered the fretting wear mechanism of Ti6Al4V alloy. In summary, the effect of shot peening on the tribological properties of Ti alloys has yet to be comprehensively studied, where limited research has reported somewhat conflicting results. In addition, no research has been published on powder metallurgical Cp-Ti alloys, even though powder metallurgy permits the manufacture of near-net-shape components, which is particularly important to produce patient-specific implants of titanium alloys [33].

To the authors' knowledge, the present study is the first to examine the effect of shot peening on the tribological properties of a powder metallurgical Cp-Ti alloy, accompanied by a detailed characterization of surface and subsurface properties such as roughness, topography, morphology, and hardness. Thus, the present study contributes to the limited literature on the influences of shot peening on the tribological behavior of titanium alloys, which could result in opening up a variety of applications for these materials, whose relatively poor tribological properties limit their use in engineering applications.

## 2. Materials and Methods

Figure 1 depicts an outline of the present study. Briefly, Cp-Ti samples were manufactured using powder metallurgy and were then shot-peened. Optical profilometry and scanning electron microscopy (SEM) were used to evaluate the surface properties of shot-peened and unpeened samples. Additionally, cross-sectional microstructures and hardness maps of shot-peened samples were examined to reveal the influences of shot

peening on microstructural and mechanical properties. Finally, ball-on-disc tests were conducted to determine the tribological properties of shot-peened Cp-Ti samples, and a detailed surface analysis of the worn samples was conducted.

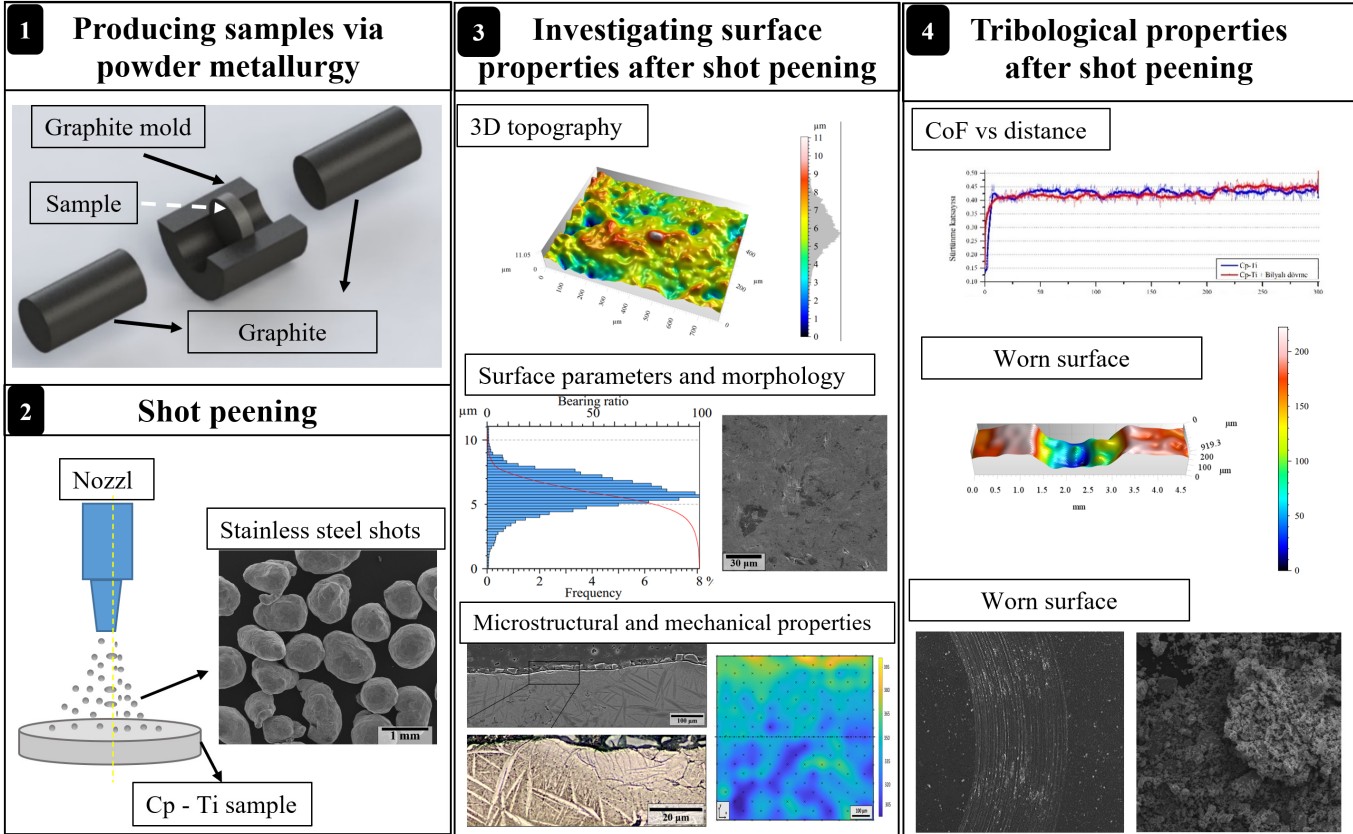

**Figure 1.** Outline of the present study.

*2.1. Processing Cp-Ti Samples*

A hot press (DIEX, China) was used to manufacture cylindrical button powder metallurgical Cp-Ti samples (height: 4 mm, diameter: 20 mm) from pure Ti powder (size range: 33 µm to 45 µm, exhibiting irregular morphology (Figure 2)). The amount of powder required was determined from the volume of the samples. In order to prevent the powder from adhering to the graphite mold during manufacturing, the mold was covered inside with graphite paper. The heights of the die punches were evenly adjusted for a homogeneous temperature distribution. To prevent powder oxidation, the sintering process was carried out in a vacuum environment (at $10^{-4}$ mbar). The samples were sintered at 950 °C under 14 kN pressure for 30 min. The formed samples were then left inside the hot press for cooling to room temperature. At the end, the samples were ground prior to shot peening using 320–1000 grits.

*2.2. Shot Peening*

The surfaces on the Cp-Ti samples were shot-peened using a specially designed fully automated shot-peening system, and shot-peening operation parameters were selected according to our previous works [14,31]. Cp-Ti samples were then peened by stainless steel shots (size range: 600 to 1000 µm (Figure 3) accelerated with compressed air (5 bar) for 150 s. Afterwards, the samples were placed in an ultrasonic bath to remove particles and dust from the sample surface due to plastic deformations on the material surface after shot peening.

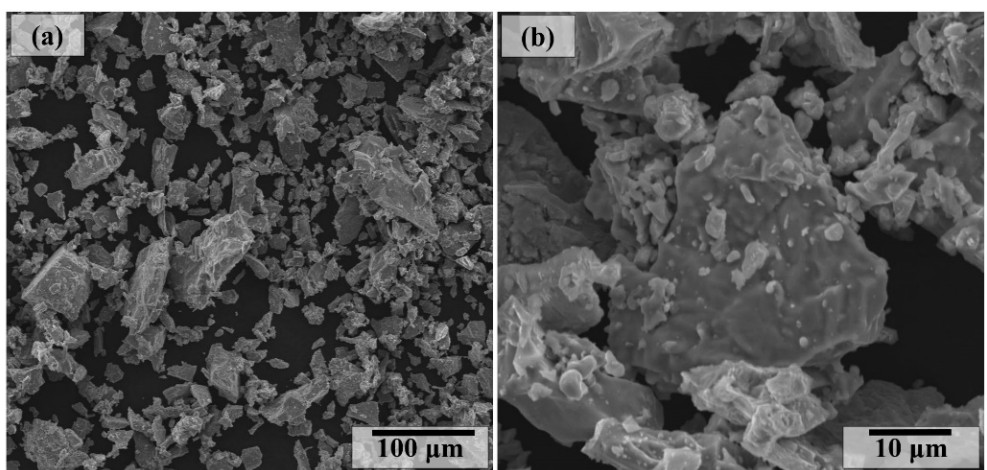

**Figure 2.** SEM images of Ti powder particles at low (**a**) and high (**b**) magnification indicating irregular morphology.

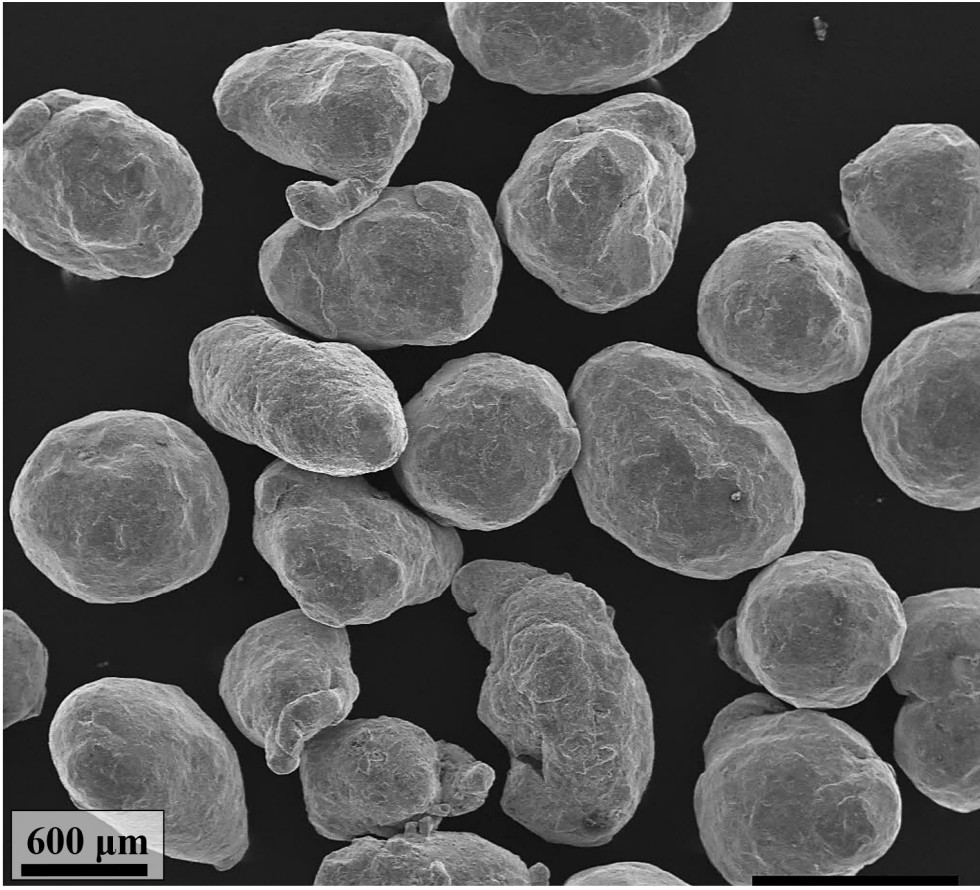

**Figure 3.** SEM images of stainless-steel shots used for peening.

### 2.3. Characterisation of Surface Properties

The 3D roughness values of the sample surfaces were determined using a 3D optical profilometer (Huvitz, Gyeonggi, Republic of Korea). Then, the 3D surface topographies, Abbott curves, surface roughness parameters, and texture isotropy have been analyzed by Mountains® 9 software (Digital Surf, Besançon, France). The surface morphologies of shot-peened samples were examined using SEM (Jeol JSM-6060, Tokyo, Japan) equipped with an energy dispersive spectroscopy (EDS) detector (Oxfords Instrument, Oxford, UK).

### 2.4. Characterisation of Mechanical Properties

The cross-sectional hardness maps of shot-peened samples were used to demonstrate the strengthening of the samples caused by shot peening. The cross-sections of shot-peened samples were metallographically prepared for hardness analysis. Briefly, they were ground (320–2000 mesh grits), polished (6-, 3- and 1-μm diamond suspension), and etched (Kroll etch: 2 mL HF, 6 mL $HNO_3$, 92 mL $H_2O$). Per ASTM E348-17, microhardness measurements were conducted using 0.2 kgf for 10 s and 0.1 mm spacing. The hardness imprints were positioned on a rectangular grid of $900 \times 540$ μm$^2$ and ~30 μm below the surface. Following this, the hardness maps were visualized by using a modified version of a customized MATLAB® code as described elsewhere [34,35].

### 2.5. Characterisation of Tribological Properties

The tribological properties of Cp-Ti and shot-peened Cp-Ti samples were examined using a ball-on-disc system. In wear tests, alumina balls (6 mm) were used as the counter surface. The samples were subjected to a normal load of 15 N over distances of 300 m and 600 m under ambient conditions, and the sliding velocity was 94.24 mm/s. Experiments were conducted in triplicates, and then CoF values were recorded by the tester. The volume loss value on the worn surface was obtained by using a 3D optical profilometer, measuring the volume loss per unit area around the wear track from 10 different regions and multiplying the wear trace circumference after taking the average of these values. This was then used to calculate the specific wear rate, *k* (1) [36]:

$$k = \frac{V}{W \times L}$$
(1)

where *V* is volume loss (mm$^3$), *W* is applied load (N), and *L* is sliding distance (m).

The surface morphological characteristics of the worn samples were investigated via SEM-EDS. Worn samples were cut from the center of the wear track, and then metallographically prepared for SEM analysis via the same procedure as described above. Additionally, the worn surfaces of the alumina balls were examined via an optical microscope. Finally, wear debris was collected after the wear tests for further SEM imaging and EDS analysis.

## 3. Results

### 3.1. Surfaces Properties of Shot-Peened Cp-Ti Alloy

The 3D surface topography, the Abbott curve, and the areal surface roughness (Sa) of the shot-peened samples are depicted below (Figures 4 and 5). As schematically illustrated elsewhere [31], the repeated impact of the shots during shot peening may cause the formation of craters due to severe plastic deformation on the surface, which results in the formation of peaks and valleys along with an increase in surface roughness. The surface topography revealed the formation of peaks and valleys as a result of the shot peening process, with a distance between the highest peak and the deepest valley around 11 μm (Figure 4a). The texture isotropy rate was 71.53%, and the angle with the highest orientation was 36°. The Abbott curves of the shot-peened surfaces revealed that the most prevalent areal roughness was 5.177 μm, occurring at a frequency of 8.1% (Figure 4b). The mean Sa of the surface was 0.999 μm, whereas the maximum peak height and depth were 5.177 μm and 5.874 μm, respectively (Figure 5).

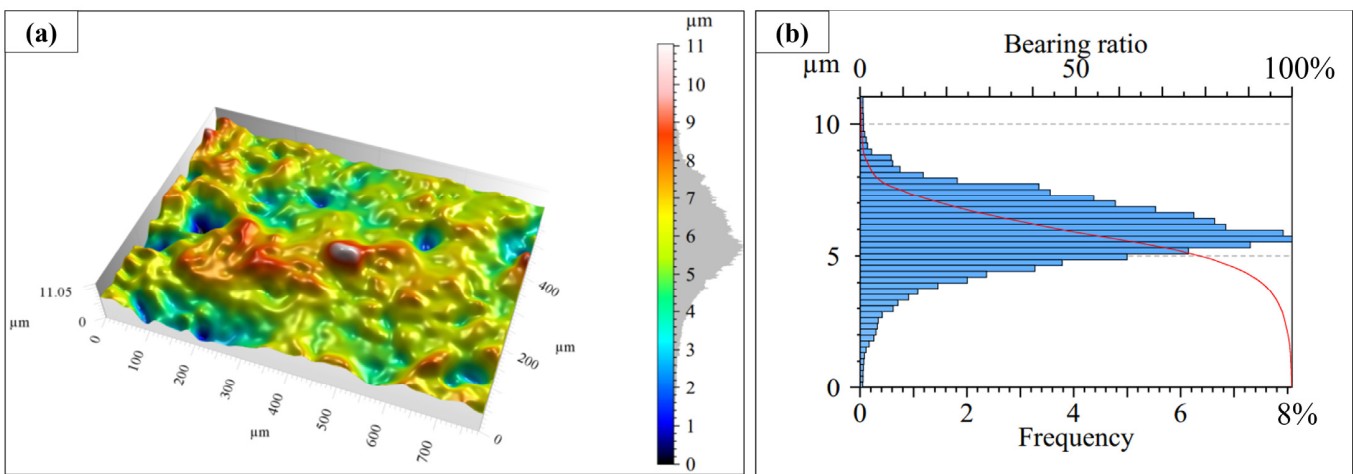

**Figure 4.** Profilometry map of shot-peened Cp-Ti surface (**a**) topography and (**b**) corresponding Abbott curve.

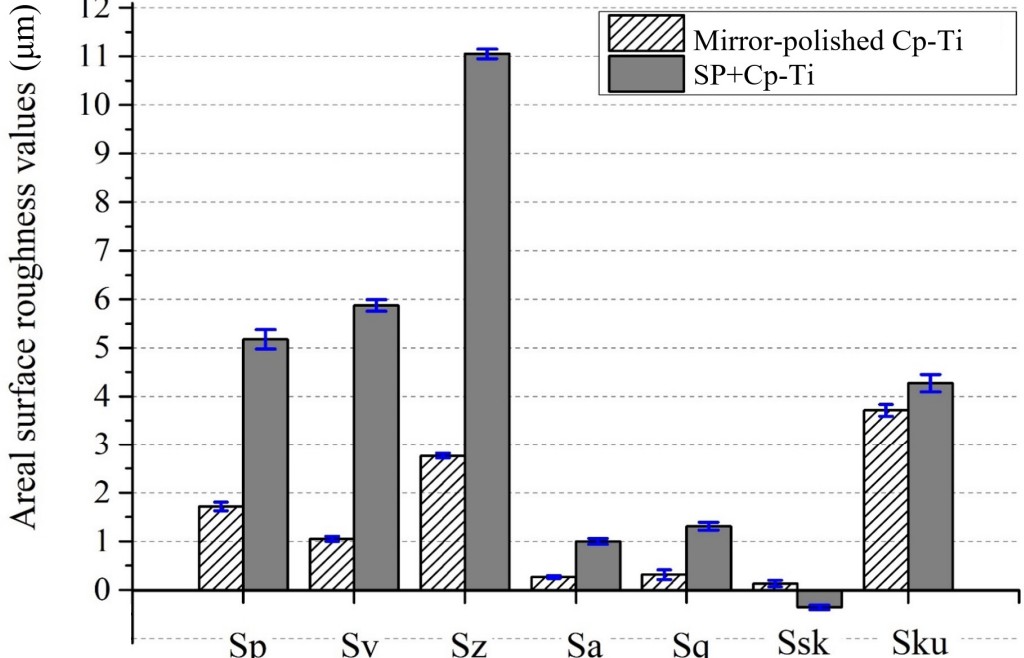

**Figure 5.** Areal surface roughness for unpeened (mirror-polished Cp-Ti) and shot-peened (SP Cp-Ti) surfaces.

The high-velocity impact of the stainless steel shots used in shot peening on the surface of Cp-Ti samples resulted in plastic deformation, micro-sliding, and micro-shearing (Figure 6), which is consistent with the increased surface roughness parameters of the shot-peened samples compared to unpeened samples (Figure 5). Micro-cracks and surface smearing were observed (Figure 6b,d, demonstrating notable plastic deformation that occurred during shot peening).

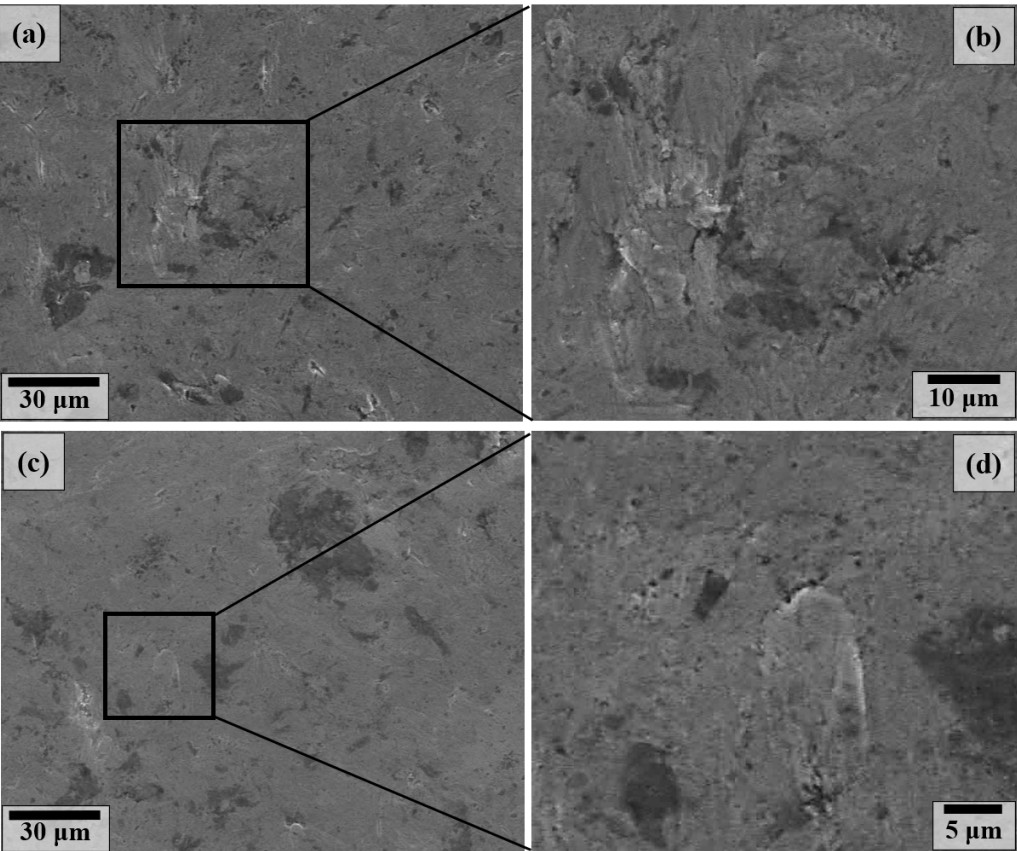

**Figure 6.** SEM images of shot-peened Cp-Ti surface at low magnification (**a**,**c**) and at high magnification (**b**,**d**).

*3.2. Tribological Properties*

The Sa of the shot-peened Cp-Ti alloy surface increased from 0.264 μm to 0.999 μm. Shot-peened and unpeened samples worn at 300 m and 600 m distances under steady conditions to determine the effect of surface modifications on the tribological behavior of Cp-Ti have yielded the following data: CoF, specific wear rate, 3D and 2D surface topography, and surface morphology, as presented in detail below.

3.2.1. Variation of Coefficient of Friction

The mean CoF curves versus sliding distance for shot-peened (SP+Cp-Ti) and unpeened (Cp) surfaces (Figure 7) illustrate two stages: a running-in stage and a steady-state stage. At a sliding distance of 300 m, the running-in stage for both shot-peened and unpeened surfaces occurred approximately within the first 10 m. In contrast, a shorter running-in stage was observed for the unpeened surfaces at a sliding distance of 600 m, whereas the running-in stage for the shot-peened surfaces ended around 10 m. During the running-in stage, CoF rapidly increased in all cases, attributable to the small contact area [37]. Once the steady-state stage was established, the CoF remained relatively stable until the end of the tests. The mean CoF for shot-peened and unpeened surfaces at a sliding distance of 300 m were 0.423 ± 0.03 and 0.425 ± 0.04, respectively. Similar values were recorded at a sliding distance of 600 m (mean CoF 0.433 ± 0.03 for shot-peened surfaces and 0.423 ± 0.05 for unpeened surfaces). These findings indicate that the effect of shot peening was limited over CoF. In addition, all surfaces showed a stable steady-state stage regardless of changes in sliding distance.

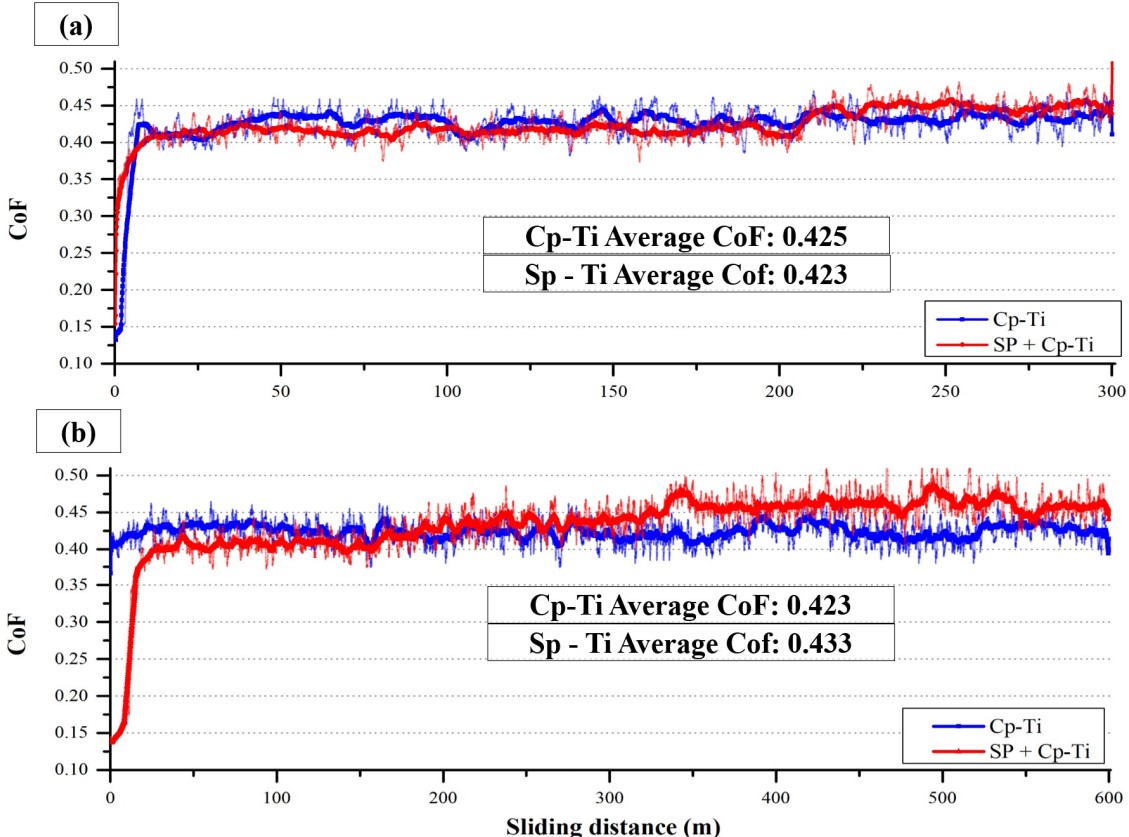

**Figure 7.** Coefficient of friction versus sliding distance of shot-peened (SP+Cp-Ti) and unpeened (Cp-Ti) at sliding distance of 300 m (**a**) and of 600 m (**b**).

### 3.2.2. Variation of Wear Rate

The specific wear rates of the shot-peened and unpeened surfaces (Figure 8) show that shot-peened surfaces had a wear rate of $450 \, (\pm \, 30) \times 10^{-6} \, \text{mm}^3/\text{Nm}$ for both sliding distances, whereas wear rates of unpeened samples were $550 \, (\pm \, 20) \times 10^{-6} \, \text{mm}^3/\text{Nm}$ and $565 \, (\pm \, 50) \times 10^{-6} \, \text{mm}^3/\text{Nm}$ at sliding distances of 300 m and 600 m, respectively. Overall, the shot-peened surfaces exhibited a lower wear rate than that of the unpeened samples, attributable to active wear mechanisms discussed later.

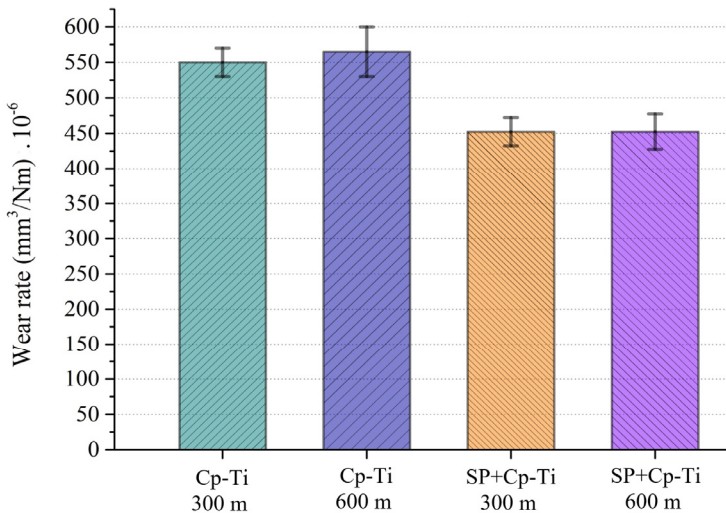

**Figure 8.** Specific wear rate of shot-peened (SP+Cp-Ti) and unpeened (Cp-Ti) at sliding distances of 300 m and 600 m.

### 3.2.3. Variation of Wear Track Topography

Figure 9 shows low resolution surface maps of the worn shot-peened and unpeened surfaces showing the width and depth of the wear track over a large region. The average wear widths of shot-peened and unpeened surfaces were 1.98 mm and 2.00 mm, respectively. The average depth of the wear track was 70 μm for shot-peened versus 80 μm for unpeened surfaces. The depth and width of the wear track were approximately the same in shot-peened surfaces, where wear depth increased in unpeened surfaces at the higher sliding distance. Peak formation around the edges of the wear track was observed on all worn surfaces and was attributable to severe plastic deformation caused by adhesive wear. To more clearly reveal the topography of the wear track, Figure 10 presents high resolution surface maps of worn unpeened and shot-peened surfaces at sliding distances of 300 m. The inhomogeneities seen at the bottom of the wear tracks (specifically, Figure 10b) indicate that abrasive wear is also prevalent during contact, most likely caused by three-body abrasion mechanisms due to the fragmentation of oxide particles and wear debris [38].

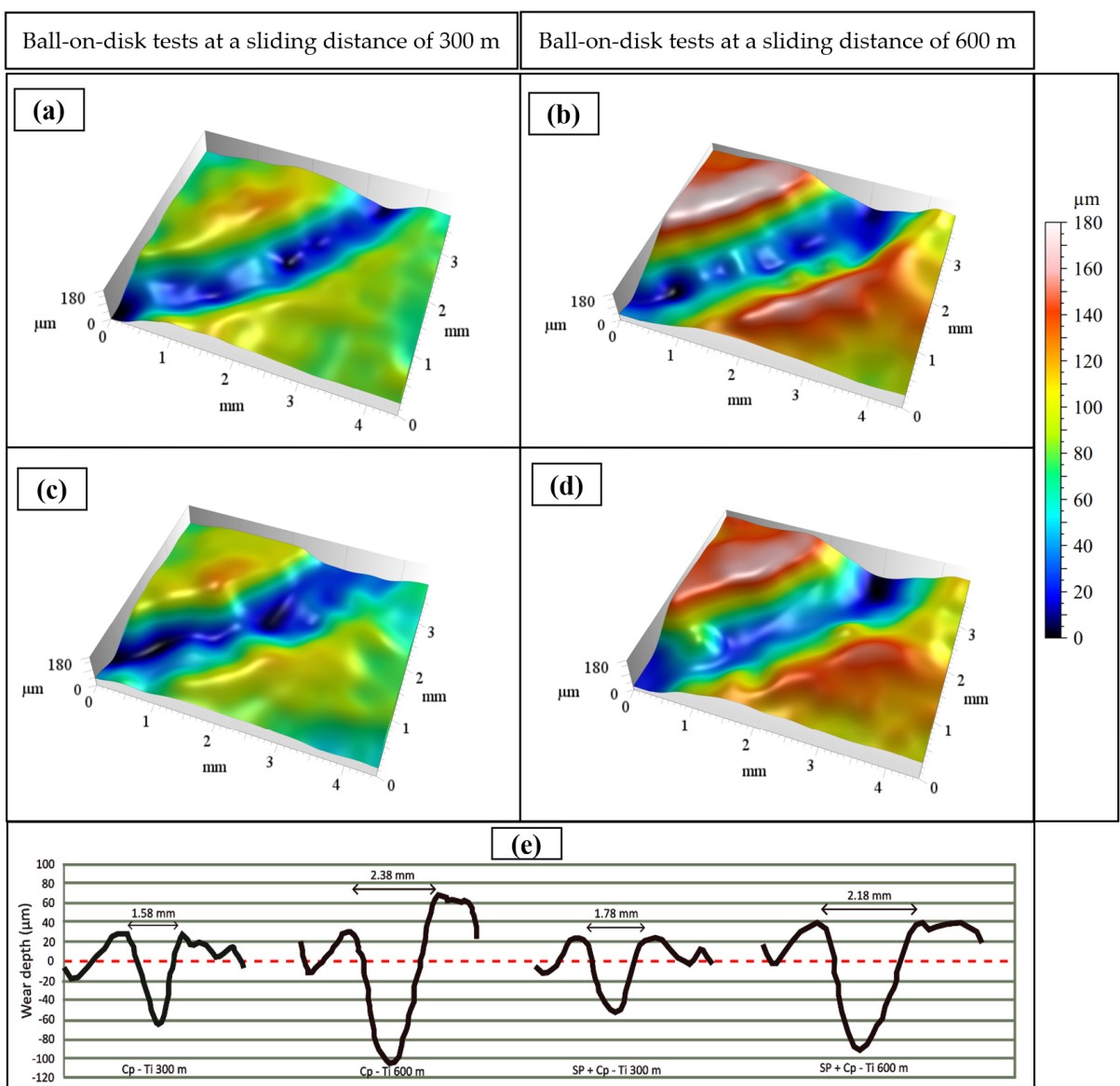

**Figure 9.** Low-resolution surface maps of (**a**,**c**) worn unpeened and (**b**,**d**) shot-peened samples at sliding distances of 300 m and 600 m and (**e**) wear depth profiles of unpeened and shot-peened samples at sliding distances of 300 m and 600 m.

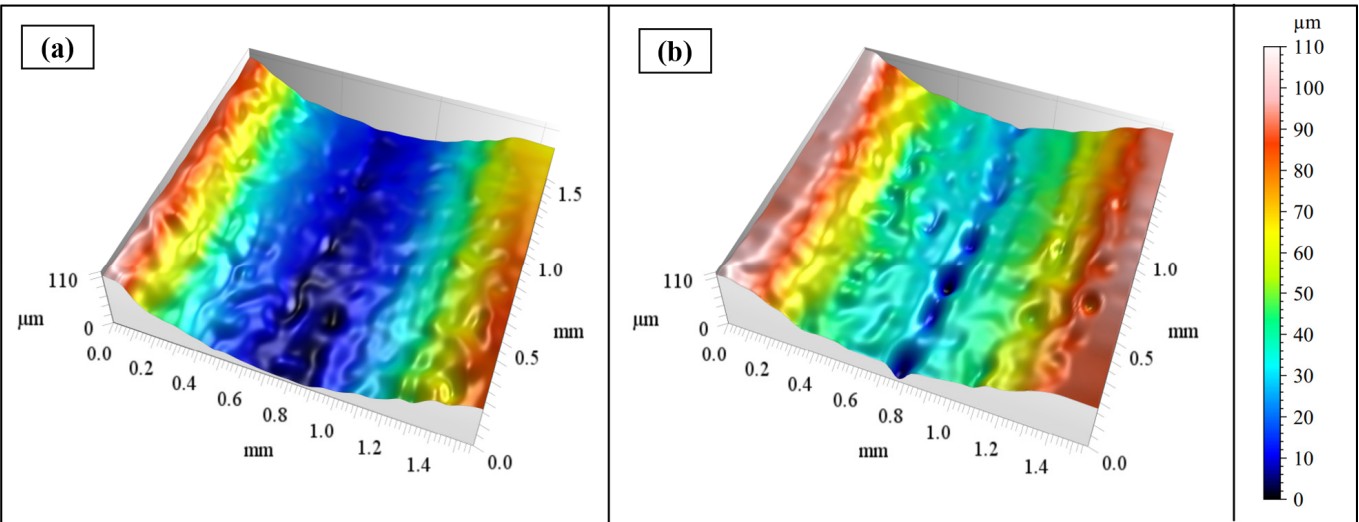

**Figure 10.** High resolution surface maps of (**a**) worn unpeened and shot-peened (**b**) samples at sliding distances of 300 m, showing decreases in width and depth of wear track after shot peening.

### 3.2.4. Worn Surface Morphology

Under the identical tribological testing conditions, the dominant wear mechanisms of the worn unpeened and shot-peened surfaces expressed certain similarities (Figure 11). Abrasion, oxidation, adhesion, and three-body abrasion have been observed on both surfaces [7,13,33]. During the contact, the surfaces underwent notable plastic deformation, leading to material removal. From the unpeened surfaces, the Ti matrix was removed mostly through micro-cutting (Figure 11a) along with signs of adhesive wear (Figure 11b). Ti-rich oxides formed a tribolayer due to friction heat on both surfaces (Figure 11c,f), confirmed by EDS analysis (Figure 12a,b). In contrast, from the shot-peened surfaces, the material was removed by ploughing without evidence of micro-cutting. More oxide islands have formed on the shot-peened surfaces, indicating a denser tribolayer formation compared to unpeened surfaces. The fragmentation of these oxide islands (Figure 11f) produced wear debris particles (Figure 11g,h); micro-scratches (Figure 11a,b) indicated particle slide and roll in the interface of the tribological system during the testing, promoting three-body abrasion [38]. The EDS spectrum of wear debris particles (Figure 12c,d) showed clear peaks of O and Ti, evidencing surface oxidation and formation of oxide islands. Figure 13 presents optical microscope images of the worn surface of the alumina ball at a sliding distance of 600 m, which indicates the transfer of material (Cp-Ti) to the alumina ball while the ball is still intact, confirming that the wear rate of the counterbody was insignificant due to the high hardness of alumina compared to Cp-Ti samples (1500 HV vs. 320 HV).

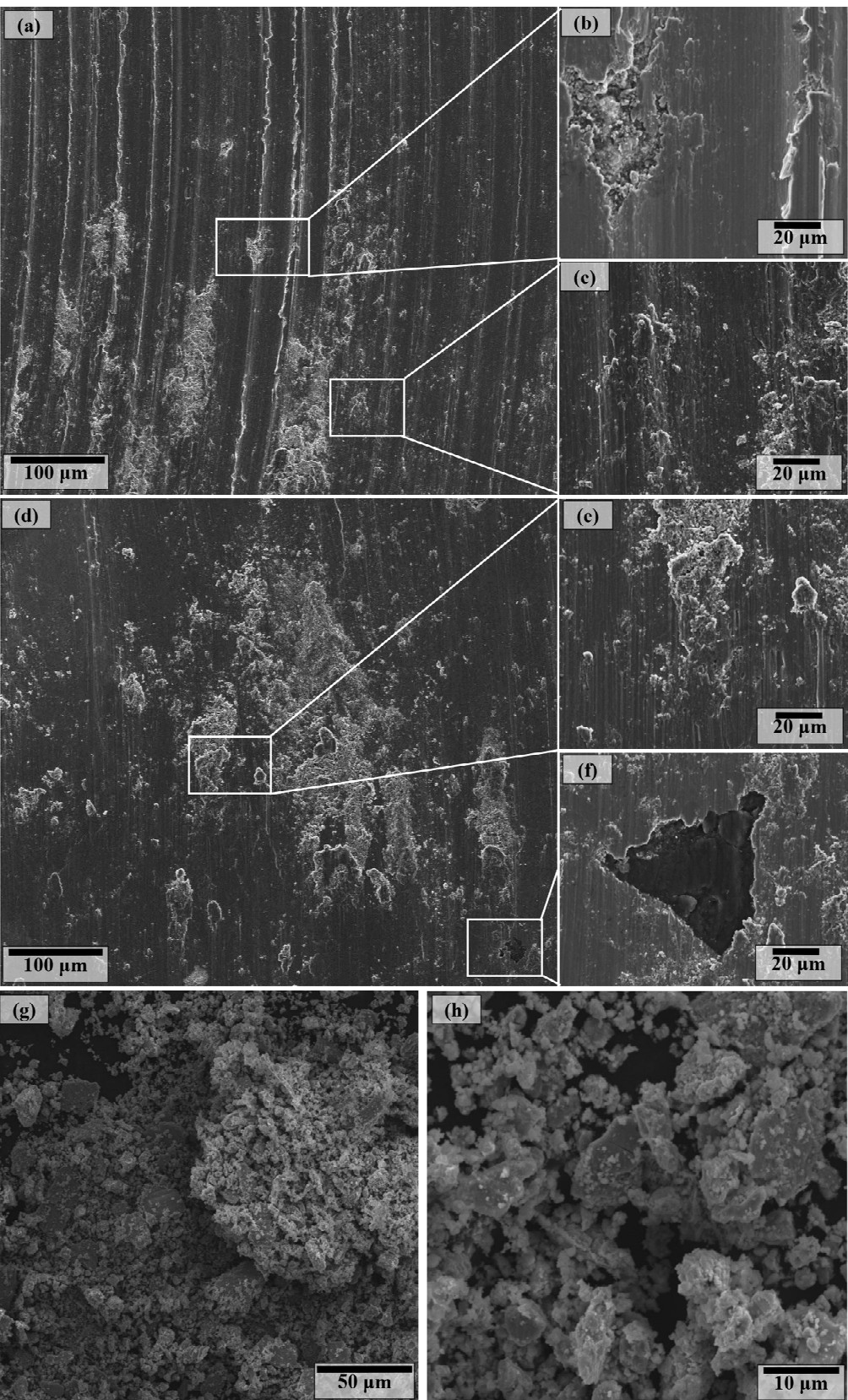

**Figure 11.** Post-test SEM of worn unpeened surface at sliding distance of 600 m at low magnification (**a**) and at high magnification (**b**,**c**); worn shot-peened surface at a sliding distance of 600 m at low magnification (**d**) and at high magnification (**e**,**f**); SEM images of wear debris particles (**g**,**h**).

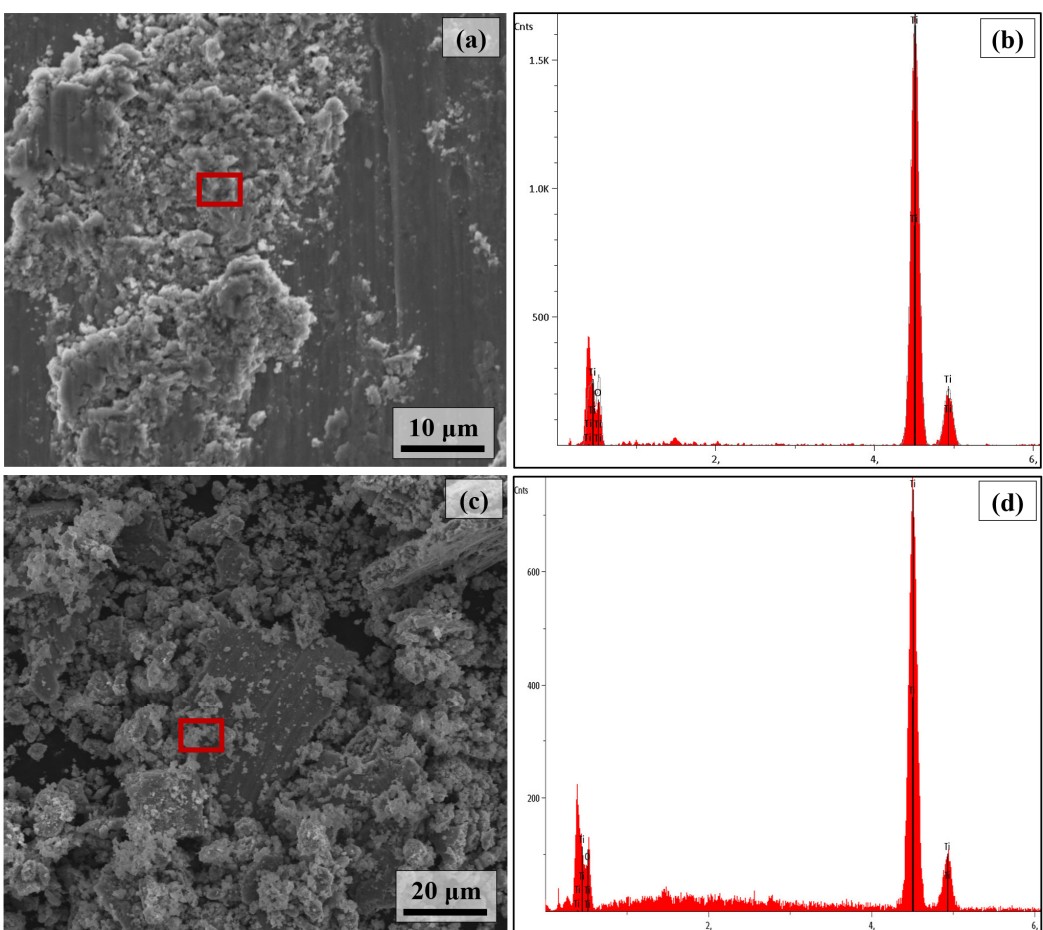

**Figure 12.** SEM image of worn shot-peened surface at a sliding distance of 600 m (**a**) and corresponding EDS analysis (**b**); SEM image of wear debris particles (**c**) and corresponding EDS analysis (**d**). Red frames show the region of interest for EDS analysis.

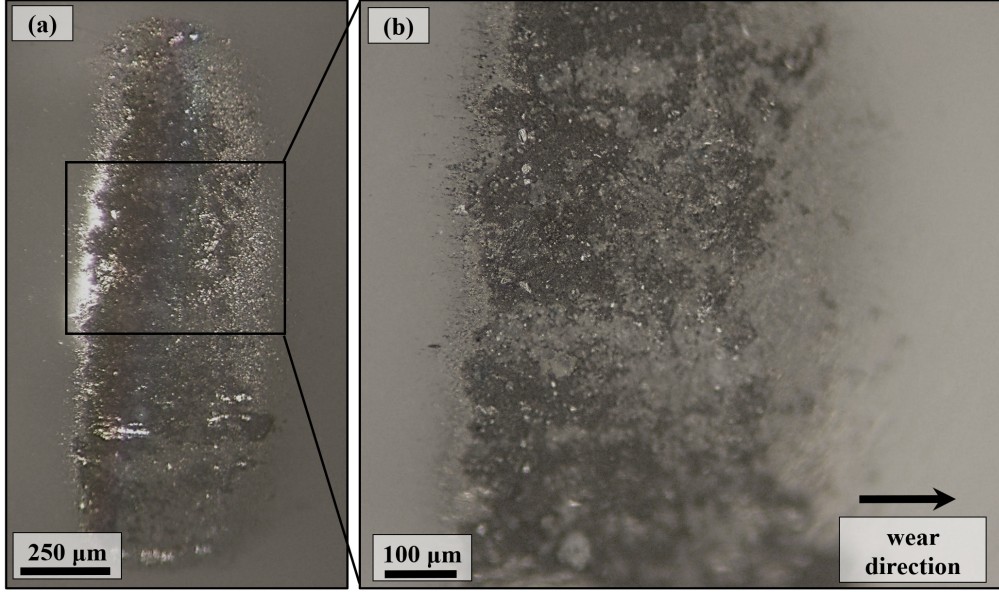

**Figure 13.** Optical microscope image of worn surface of the alumina ball at a sliding distance of 600 m at low magnification (**a**) and at high magnification (**b**), indicating the transfer of material (Cp-Ti) to the alumina ball while the ball was still intact.

## 4. Discussion

Shot peening notably altered the surface properties (roughness, topography, and morphology) of Cp-Ti samples (Figures 4–6). The plastic deformation induced by the impact of shots modified the surface properties via various deformation mechanisms (e.g., folding, removal of ridges, crater formation), as illustrated and discussed previously [31]. These changes not only resulted in increased surface roughness (Figure 5) but also altered the microstructural and mechanical properties of the surface and subsurface of shot-peened Cp-Ti samples as discussed below. These modifications further played a role in the tribological behavior of the alloy and were expected to be influenced by changes in surface topography, roughness, microstructure, and mechanical properties.

Results from the studies examining the effects of shot peening on the tribological behavior of materials were compared to those from the present study (i.e., percent change in wear rate, Figure 14). Briefly, in the present study, shot peening resulted in around 20% reduction in the wear rate, in accordance with the majority of the literature reporting that shot peening reduces it (as high as 90%), along with a few exceptions mentioning increases (e.g., [17,21]). For the moment, based on the literature, it is not possible to confidently generalize about the effect of shot peening on the tribological performance of alloys. This is primarily due to the fact that both shot peening and tribological testing are largely dependent on selected materials, equipment, and test parameters. It should be noted that various materials may exhibit different deformation behavior under shot peening based on their intrinsic properties (e.g., crystal structure, plasticity); therefore, it is essential to investigate the underlying causes of changes in wear resistance in order to fully comprehend the relationship between shot peening and tribological behavior for each material system. Another point to highlight is that the literature has focused primarily on steel alloys, whereas studies on Ti alloys are still limited [17,26,32]. Regarding these studies on Ti alloys, conflicting results have been reported thus far [17,32]. Zebrowski and Walczak [32] observed a positive effect of peening on the tribological behavior of Ti6Al4V alloy, which was attributable to the strain-hardened surface layer of the alloy with shot peening. In contrast, DiCecco et al. [17] concluded that shot peening had no effect on the wear rate of alpha titanium alloy, including TiB particles, but they hypothesized that shot peening affected the wear mechanisms.

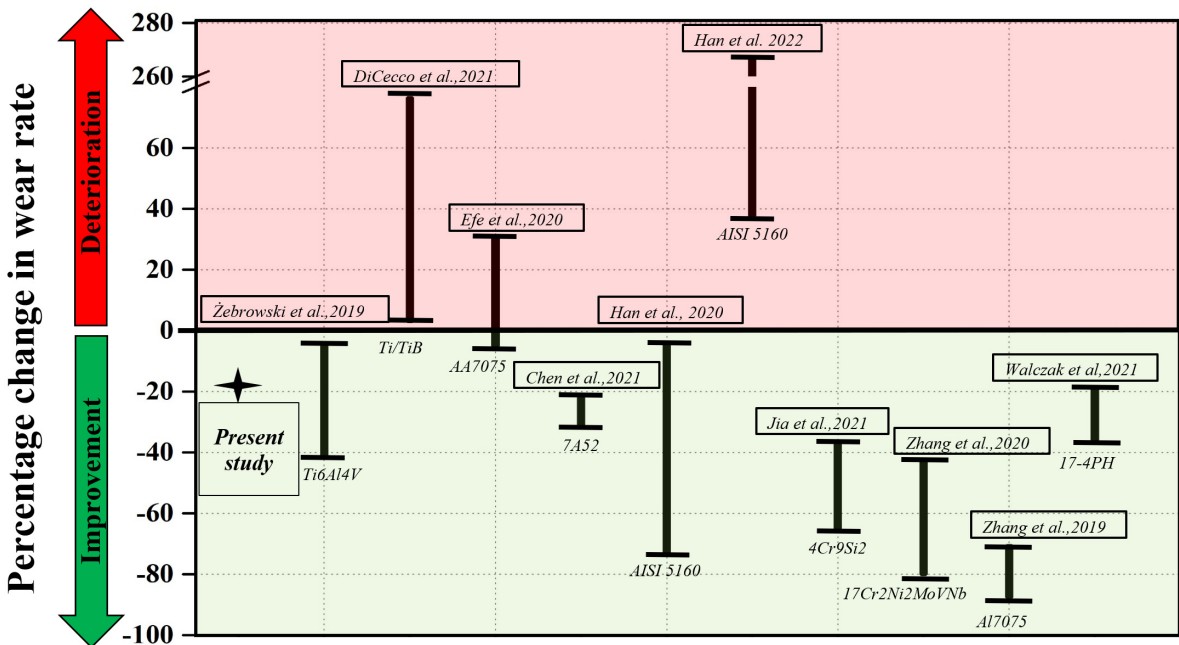

**Figure 14.** Comparison of percentage change in wear rate with literature [16–18,20–22,25,29,30,32] (details in Supplementary Materials).

The overall influence of shot peening on CoF was minor (Figure 7), where values for unpeened and shot-peened samples ranged from 0.423 to 0.433. More specifically, the CoF for shot-peened samples remained slightly below the CoF of unpeened samples up to around 200 m, after which it passed the CoF of unpeened samples and then remained marginally above it. The CoF of Cp-Ti manufactured at a similar sintering temperature and tested under comparable tribological conditions has been reported in the range of 0.45 and 0.58 [1,3,7,33]. The CoF for the unpeened surfaces of Cp-Ti in the present study corresponds to the lower end of this range. Regarding the CoF of shot-peened samples, conflicting results were reported in the literature: The majority of the research has reported a decrease in CoF after shot peening [16,22,28–30,32], whereas a few studies noted a small increase [17,18,25]. Those reporting a decrease attributed the reduction after shot peening to the following factors: formation of an oxide film that improves lubrication and reduces friction [22], improved mechanical properties such as microhardness [30,32], and/or high compressive residual stress obtained by shot peening [30]. In contrast, increases in CoF were primarily attributed to elevated surface roughness and topographical parameters after shot peening [25]. In the present study, while shot peening caused important changes in topographical parameters (Sections 3.1 and 4) as well as surface/subsurface hardness and microstructural features (discussed below), CoF remained mostly constant. This is possibly due to the combined effect of these modifications having led to a resulting consistent CoF.

After shot peening, the wear rate for the samples decreased by around 20% (Figure 8), resulting in improved wear resistance of Cp-Ti samples. SEM and optical microscope images of the cross-sectional microstructures of the shot-peened and unpeened samples after 600 m of wear would provide a better understanding of the shot-peening-induced improvement in wear behavior (Figures 15 and 16). Specifically, the cross-sectional microstructures of the wear track reveal that unpeened material has a higher plastic deformation, resulting in rapid material removal. The wear track of the shot-peened sample was observed to be flatter than that of the unpeened sample, indicating that the active wear mechanisms differ as a result of the formation of a strain-hardened surface/subsurface layer and the modification of the microstructure due to the plastic deformation induced by shot peening (Figures 15 and 16). Micro-cutting was more prevalent in shot-peened samples, whereas micro-ploughing was predominant in unpeened samples, as evidenced by the waviness of the surface. In addition, vertical and horizontal microcracks are visible just beneath the wear track (Figure 13), indicating a clear sign of plastic deformation caused by compressive and shear stresses induced by the counter body during the contact [34]. Consequently, unpeened Cp-Ti samples undergo wear by micro-ploughing and micro-cracking because of excessive plastic deformation during wear. In contrast, the flattened wear track surface and limited crack formation beneath the wear track of shot-peened material suggest that the hardened surface/subsurface layer (Figure 15) better accommodated plastic deformation and underwent wear via micro-cutting, thereby reducing material removal.

The peened material has a work-hardened layer (as evidenced by hardness mapping, Figure 17) that slowed down material removal and plastic deformation during contact. In addition, optical and SEM cross-sectional images of shot-peened Cp-Ti samples prior to wear testing reveal the surface deformation and modification of subsurface microstructural features (Figure 18). Since the sintering temperature (950 °C) was above β transus, the resulting microstructure consists primarily of acicular α′ martensite and coarse elongated α grains due to the insufficient cooling rate during the β → α transformation (Figure 18a). Although an ultrafine-grained layer (i.e., nanostructured gradient layer) is formed just beneath the surface down to a depth around 20 μm (Figure 18a), it is not possible to clearly visualize and determine the average grain size of this layer due to the insufficient resolution of the microscopy techniques used. However, the cross-sectional hardness map of shot-peened Cp-Ti samples (Figure 17) reveals a sharp increase in hardness in this region, indicating that shot peening significantly modifies the near-surface microstructure, resulting in an increase in wear resistance, as previously demonstrated (Figure 14). To the authors' knowledge, the present study is the first to demonstrate the increase in hardness

of shot-peened materials in 2D via hardness mapping, thereby elucidating why the wear resistance of shot-peened materials is superior to that of unpeened materials.

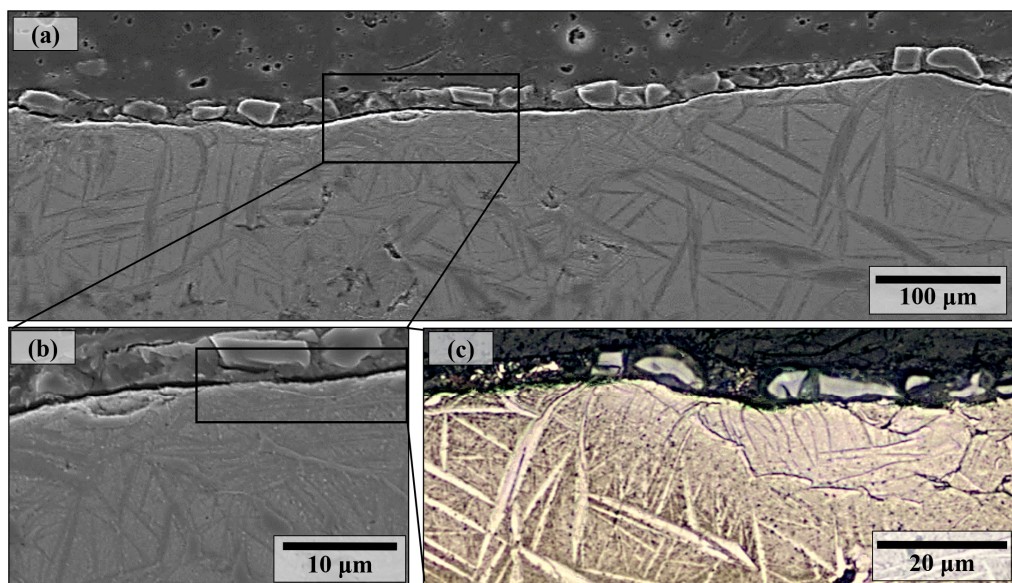

**Figure 15.** Cross-sectional SEM (**a**,**b**) and optical images (**c**) of wear affected zone in unpeened Cp-Ti samples at sliding distance of 600 m.

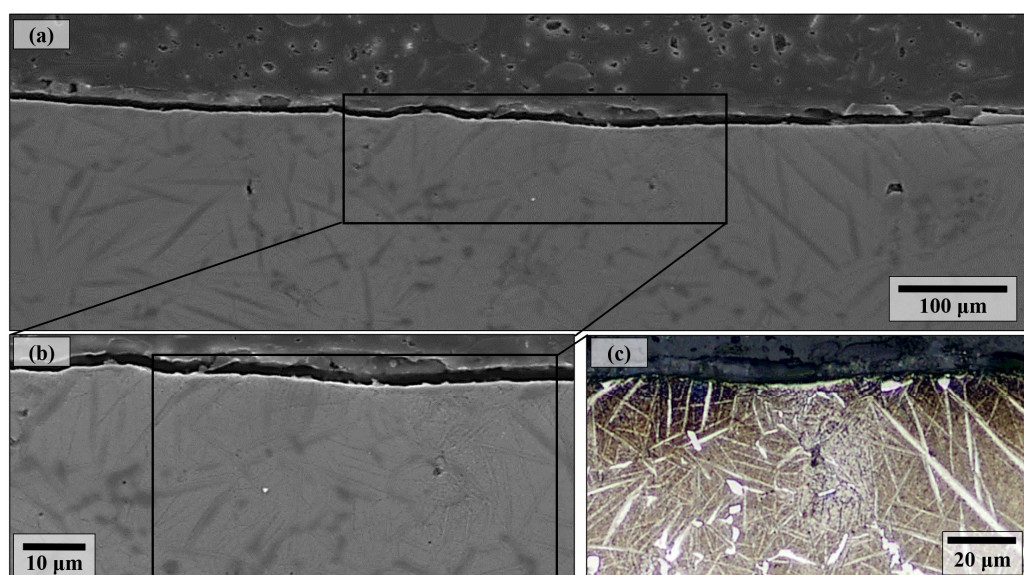

**Figure 16.** Cross-sectional SEM (**a**,**b**) and optical images (**c**) of wear affected zone in shot-peened Cp-Ti samples at sliding distance of 600 m.Cross-sectional SEM (**a**,**b**) and optical images (**c**) of wear affected zone in shot-peened Cp-Ti samples at sliding distance of 600 m.

CoF, which is a ratio of the frictional force between two surfaces, is fundamentally associated with the physical nature of the materials in contact (e.g., surface chemistry, topography) [22], whereas the wear rate (e.g., wear resistance) is strongly related to their microstructural features (grain size, phases) and mechanical properties (e.g., hardness, plasticity, toughness, resilience). Shot peening is likely to affect CoF because it modifies the surface topography (specifically roughness), which can influence CoF [25]. In spite of this, shot peening has no noticeable impact on the CoF of Cp-Ti in the present study, whereas it has modified the mechanical properties of the alloy (i.e., hardness) and the microstructural features, resulting in an increase in wear resistance (i.e., decrease in wear rate) with shot peening.

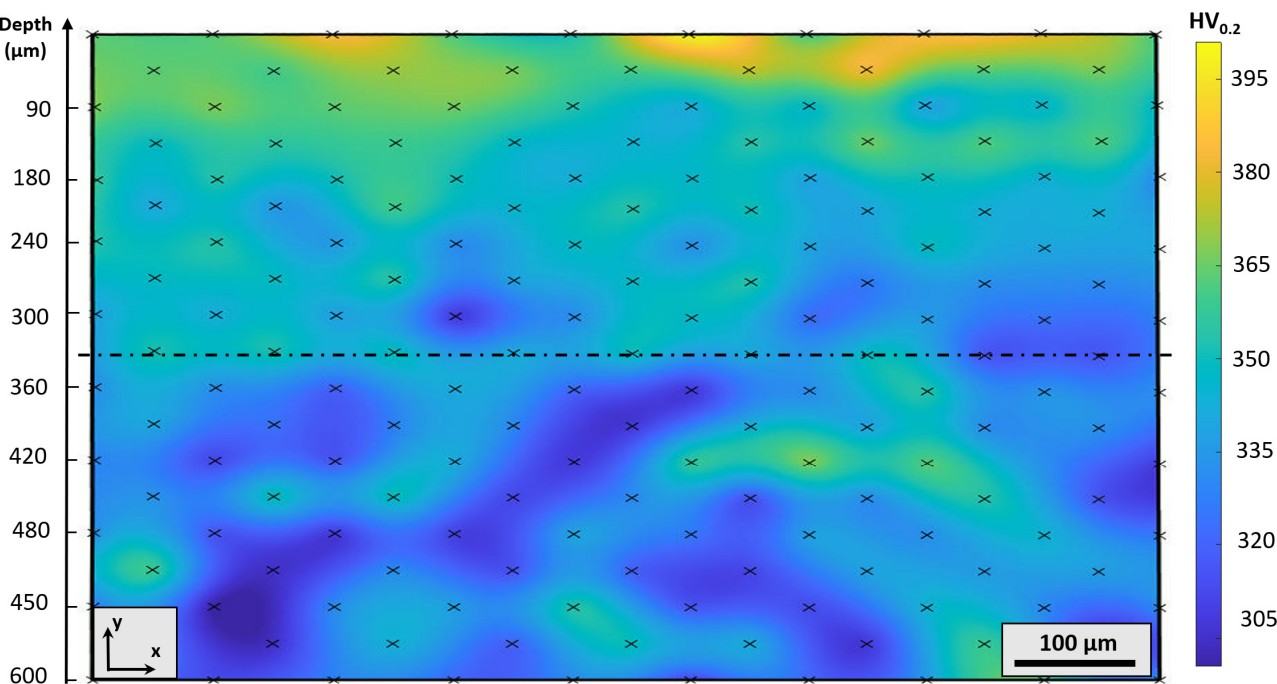

**Figure 17.** Cross-sectional hardness map of shot-peened Cp-Ti sample, indicating surface/subsurface hardening of the material with shot peening. X marks indicate the location of hardness impressions, and dashed line shows the depth of hardened layer, which is around 330 μm thick.

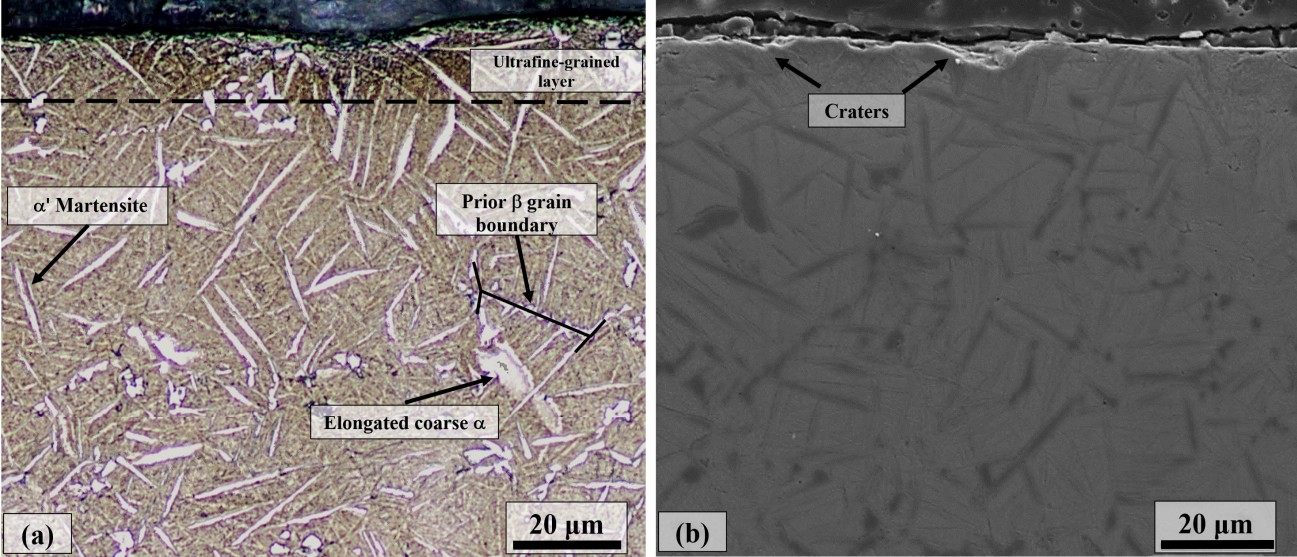

**Figure 18.** Cross-sectional optical (**a**) and SEM (**b**) images of shot-peened Cp-Ti sample, indicating deformation of surface and modification of subsurface microstructural features.

## 5. Conclusions

Relatively poor wear resistance of Ti alloys and contradictory results regarding their tribological behavior limit their application in the engineering domain. A thorough assessment of the effect of cold working processes such as shot peening on the tribological properties of Ti alloys is thus needed. To the authors' knowledge, this is the first study to investigate the effect of shot peening on the tribological properties of a commercially pure titanium (Cp-Ti) alloy. Cp-Ti samples were manufactured via a cost-effective powder metallurgy process, followed by shot peening via a custom-designed peening system. A comparison of the surface properties of shot-peened and unpeened samples was performed

via optical profilometry and SEM-EDS, and the tribological properties of Cp-Ti samples and surface analysis of the worn samples were carried out.

- The application of shot peening led to notable plastic deformation of the Cp-Ti alloy, resulting in the formation of craters, peaks, and valleys; and increased surface roughness that may affect tribological behavior. In addition, shot peening modified the microstructural features near the surface, resulting in the formation of a thin ultrafine-grained layer beneath the surface. However, it was not possible to clearly visualize and determine the average grain size of this layer due to the insufficient resolution of the microscopy techniques used. Therefore, the authors suggest that future research should focus solely on implementing advanced electron microscopy techniques (i.e., high-resolution SEM and electron backscatter diffraction) to reveal the nano-gradient and ultrafine-grained microstructural features after shot peening.
- Cross-sectional hardness mapping of shot-peening samples was used for the first time to reveal the formation of a work-hardened surface layer with shot peening, where a sharp increase in hardness was observed within the near surface, indicating that shot peening significantly modifies the near-surface microstructure.
- The coefficient of friction was similar for both shot-peened and unpeened samples, and changes in the sliding distance did not have a substantial effect.
- High-resolution 3D surface topographies of worn unpeened and shot-peened surfaces revealed micro-scratches and inhomogeneities along wear tracks, which were indicative of three-body abrasion mechanisms during contact.
- Ploughing was the primary mechanism of material removal in the shot-peened samples with no evidence of micro-cutting. EDS revealed the formation of oxide islands, suggesting denser tribolayer.
- Shot peening successfully reduced the wear rate of Cp-Ti samples due to the formation of a work-hardened layer and the modification of the microstructural features, although it is essential to further investigate the relationship between the modified microstructural features, mechanical properties and wear resistance of shot-peened materials.

**Supplementary Materials:** The following supporting information can be downloaded at: https://www.mdpi.com/article/10.3390/coatings13010089/s1, Table S1. Wear rate improvement of the present study and literature studies.

**Author Contributions:** Conceptualization, Y.Y.A., E.I. and E.A. (Egemen Avcu); methodology, Y.Y.A., E.I., A.Ö., E.Ç., E.A. (Eray Abakay), F.G.K., R.Y., A.C. and E.A. (Egemen Avcu); software, Y.Y.A., E.I. and E.A. (Egemen Avcu); validation, Y.Y.A., E.I., E.A. (Eray Abakay), M.G., F.S. and E.A. (Egemen Avcu); formal analysis, Y.Y.A., E.I., E.A. (Eray Abakay), and E.A. (Egemen Avcu); investigation, Y.Y.A., E.I., A.Ö., E.Ç., E.A. (Eray Abakay), M.G. and E.A. (Egemen Avcu); resources, E.A. (Egemen Avcu), and M.G.; data curation, Y.Y.A., E.I., A.Ö., E.Ç., E.A. (Eray Abakay), F.G.K., R.Y., A.C., M.G. and E.A. (Egemen Avcu); writing—original draft preparation, Y.Y.A., E.I., M.G. and E.A. (Egemen Avcu); writing—review and editing, Y.Y.A., E.I., A.Ö., E.Ç., E.A. (Eray Abakay), F.G.K., R.Y., A.C., F.S., M.G. and E.A. (Egemen Avcu); visualization, Y.Y.A., E.I., F.S. and E.A. (Egemen Avcu); supervision, Y.Y.A., E.I., M.G. and E.A. (Egemen Avcu); project administration, E.A. (Egemen Avcu) and M.G.; funding acquisition, E.A. (Egemen Avcu) and M.G. All authors have read and agreed to the published version of the manuscript.

**Funding:** The authors acknowledge the financial support by Kocaeli University Scientific Research Projects Coordination Unit (Project Number: 2503).

**Institutional Review Board Statement:** Not applicable.

**Informed Consent Statement:** Not applicable.

**Data Availability Statement:** Not applicable.

**Conflicts of Interest:** The authors declare that they have no conflicts of interest.

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
