# Peer review of "Surface and Tribological Properties of Powder Metallurgical Cp-Ti Titanium Alloy Modified by Shot Peening"

_coatings, doi:10.3390/coatings13010089_

Round 1

Reviewer 1 Report

I'm very glad to review the paper in greater depth. This paper investigated the effect of shot peening on the wear resistance of commercially pure titanium in powder metallurgy. However, some problems in this paper make me have to suggest an overhaul. Therefore, the major revision needs to be implemented before publication. The main problems are summarized as follows:

(1) The author mentions that "For the first time, shot peening was used for mechanical surface treatment to modify the surface and tribological properties of a powder metallurgical pure titanium alloy (CP-Ti). " However, the authors do not express the differences between Cp-Ti prepared by powder metallurgy and Cp-Ti prepared by conventional methods. Therefore, this paper is still concerned with the effect of shot peening on the mechanical properties of Cp-Ti, which has been the subject of many studies, and the SMAT method is very similar to shot peening. This reflects a very weak motivation for the study. The authors are requested to make the appropriate changes.

(2) Keywords: "mechanical surface treatment, plastic deformation, surface engineering" cannot be used as keywords due to their very broad nature. The addition of "shot peening" is suggested.

(3) Introduction: This section is too much and should be shortened. In particular, the second and third paragraphs. Furthermore, I believe that the SMAT method and shot peening are identical. Both methods are designed to form plastic deformation layers on surfaces and sub-surfaces to increase surface mechanical properties. Only the defects obtained by the SMAT method are more numerous. The authors should clarify this point rather than separating the two methods.

(4) Results: In fact, only one type of shot peening process was used in this paper, so this study lacks systematicity; Figures 10~12 can be combined; I think SEM images of specimen cross-sections before and after shot peening should be given; These data can be compared with the SEM images after frictional wear to obtain microstructure evolution patterns.

Author Response

Please see attached detailed response letter.

Reviewer 2 Report

This manuscript introduce the shot peening to performe the mechanical surface treatment to modify the surface and tribological properties of a powder metallurgical pure titanium alloy (Cp-Ti). It is an interestng work for the researchers. The structural charactierization  is fully. I recommend to accept it for publication as it is. 

Author Response

Response to comments of Reviewer #2:

This manuscript introduce the shot peening to performe the mechanical surface treatment to modify the surface and tribological properties of a powder metallurgical pure titanium alloy (Cp-Ti). It is an interestng work for the researchers. The structural charactierization is fully. I recommend to accept it for publication as it is.

Response: We sincerely appreciate kind comment of the reviewer.

Reviewer 3 Report

The authors studied the "Tailoring Surface and Tribological Properties of Powder Metallurgical Cp-Ti Alloy via Shot Peening". The research work is good and can be accepted for publication with minor revision.

1. The conclusion should be given point by point showing the important findings.

2. The microstructural features should be pointed out by arrows in the micrographs in all microstructures. 

3. The research gap should be given in introduction section.

Author Response

(The authors gave the same response as above.)

Reviewer 4 Report

The current work presents the influence of shot pinning on the tribological properties of Cp.Ti. The authors have presented the results and the observations in detail. However, the below issues need to be addressed in the revised manuscript.

1. The initial microstructures before the wear test of the shot-pinned and normal samples are not clear. The depth of the shot pinned affected region needs to be shown and measured. Shot pinning results in gradient microstructures from the top surface to the core; authors need to establish the same. Further, these gradient microstructures result in gradient hardness- which also needs to be established.  

2. The COF, mode of wear, and the worn surface are similar in the shot-pinned and normal samples. Then why the wear rate is different?

3. The shot pinning process parameters significantly affect the surface hardness and the depth of the shot pin-affected region. Since the wear rate improvement presented in this work is not significant, how did the authors arrive at those shot-pinning conditions, or did they try other conditions?  

Author Response

(The authors gave the same response as above.)

Reviewer 5 Report

Commentary and comments are in the appendix.

Author Response

(The authors gave the same response as above.)
